# Enhancing Wire-Rope Damage Signals Based on a Radial Magnetic Concentrator Bridge Circuit

**DOI:** 10.3390/s22103654

**Published:** 2022-05-11

**Authors:** Jie Tian, Wei Wang, Hongyao Wang, Qiang Bai, Zeyang Zhou, Pengbo Li

**Affiliations:** 1School of Mechanical Electronic and Information Engineering, China University of Mining & Technology, Beijing 100083, China; tianj@cumtb.edu.cn (J.T.); 17862666175@163.com (W.W.); baiqiang1997@163.com (Q.B.); 15600209322@139.com (Z.Z.); 17349808206@163.com (P.L.); 2Key Laboratory of Intelligent Mining and Robotics, Ministry of Emergency Management, Beijing 100083, China

**Keywords:** finite element analysis, magnetic concentrator bridge, radial magnetic vector, signal characteristic, signal extraction, wire-rope detection, flexible printed circuit

## Abstract

Hall-effect sensors are used for the non-destructive testing of wire ropes owing to their low power consumption and high operation frequency. The high-speed operation of wire ropes causes vibration inclination at different frequencies, which makes it difficult to detect the ropes. Considering that the radial signal in the magnetic flux leakage (MFL) detection method can respond to damages to the maximum extent possible, this study proposes a radial magnetic concentrator suitable for the non-destructive testing of wire ropes based on theoretical analysis and transient magnetic field simulations. The concentrator improves the radial magnetic circuit, polymerizes the leakage of the magnetic field in the detection device, and the leakage of the magnetic field of the defect converges at the sensor position of the circumferential array to improve the signal-to-noise ratio of the Hall-effect sensor. In addition, the MFL field is homogenized through the structure of the magnetic concentrator when the wire rope is tilted, which weakens the influence of the vibration tilt of the wire rope on the test results. Finally, the experiments show that the amplitude of the wire-rope damage signal is effectively improved by using the proposed radial magnetic concentration technology, hence being convenient for defect analyses.

## 1. Introduction

Wire ropes are usually used in the continuous vibration environment, but may be broken, or exposed to wear, rust, and other damages [1]. Wire ropes are usually twisted from high-quality carbon structural steel wires, which are typical ferromagnetic materials. The magnetic flux leakage (MFL) detection technique is extensively used to diagnose the surface and internal defects of ferromagnetic parts such as wire ropes, oil and gas pipelines, and tracks, because of its advantages associated with a lack of pollution, lack of coupling agents, and its high reliability [2,3,4]. The vibration load on the part not only affects the structural stability but also interferes with leakage detection [5,6].

To solve the interference caused by the wire rope in the process of vibration, many scholars at home and abroad have made various improvements to the leakage detection method, excitation detection device, and leakage signal processing to reduce the impact of vibration on leakage detection. For example, Kaczmarczyk in Germany designed an electromagnetic sensor based on Hall-effect sensors by arranging 30 sensors in the circumferential direction of the wire rope to establish a three-dimensional signal associated with the rope’s damage, thus observing the distribution of defective damage in the axial and circumferential directions of the wire rope [7]. Wang et al. studied the effects of the lift-off value on the Hall-effect sensor and designed a magnetic circuit structure suitable for the Hall-effect sensor according to the variation law of the lift-off value that can ensure that the Hall-effect sensor maintains high sensitivity in detecting the wire rope [8]. Zhang et al. designed a magnetic concentrator sensor, which allows full acquisition of the MFL with a magnetic centralizer, and the number of Hall-effect sensors used was reduced; this simplified the subsequent signal processing [9]. Yu et al. used a combination of finite elements and genetic algorithms to optimize the excitation source and leakage distribution that improved the reliability of wire-rope detection [10]. Scholars have also used various signal processing methods to reduce vibration noise. For example, Liu et al. used a combined signal processing method based on a trap filter and wavelet denoising [11]. Zhao et al. used a two-dimensional Fourier transform approach to reduce vibration noise [12].

However, increasing the number of Hall-effect sensors through the ring array makes the signal processing more burdensome, and the subsequent eigenvalue extraction becomes cumbersome. The magnetoresistive sensors are used to improve the sensitivity of the sensor while lift-off values need to be operated within a very small variation range, which is difficult to apply in online wire-rope detection [13]. The rapid development of magnetic concentrating detection technology in the field of inspection [14,15] provides a new way of development to solve the wire-rope vibration. The principle of magnetic concentrating detection is achieved by adding ferromagnetic components to the air gap, thus improving the air-gap magnetic field in order to achieve wire-rope detection by a small number of magnetic sensitive elements [16]. Wang et al. [15] analyzed the performance of the magnetic concentrator to collect MFL by finite element simulation and proposed a structure suitable for the axial collection of leakage flux.

In the above magnetic concentration technique, the concentrator mainly acts as a homogenizer for the leakage magnetization parallel to the main magnetic flux. However, Hall-effect sensors for radial detection have smaller lift-off values, which is more beneficial for wire rope non-destructive testing. The radial magnetic concentrator provides the Hall-effect sensor with the ability to maintain a small lift-off value while acting as a convergent magnetic field.

In this study, a sensor consisting of a radial magnetic concentrator and a Hall-effect sensor array is designed to detect wire-rope damage. The structural parameters of the magnetic concentrator were optimized by the finite element method, and the relationship between the magnetic bridge and air gap of the radial magnetic concentrator and the performance of the magnetic concentrator collecting MFL were investigated. In addition, considering the effect of wire-rope tilting on radial leakage magnetic acquisition, we also studied the variation of leakage magnetic field when the wire rope was tilted following the installation of a radial magnetic concentrator. Finally, the sensor was applied to the wire breakage detection experiment, and the waveform was plotted in MATLAB; in this way, the MFL signal could be identified clearly.

The remainder of this article is organized as follows. In Section 2, the principle of wire-rope leakage scanning is introduced and the two-dimensional magnetic dipole model when the wire rope is tilted is analyzed. In Section 3, the mathematical and simulation models of the radial magnetic concentrator are described in detail. In Section 4, the design of sensors and experimental circuits in the study is presented. Experiments, steps, and the analysis of results are given in Section 5. Section 6 concludes the article.

## 2. Magnetic Field Modeling and Analysis

The principle of radial MFL detection is shown in Figure 1. The wire rope is magnetized to saturation by the permanent magnet in which a soft magnetic material is used as the armature in its peripheral circuit. The connect iron acts as fixed support for the permanent magnet and the conductive magnetic field and acts as a closed magnetic circuit between the external magnetic bridge and the wire rope. When the damaged wire rope passes through the flaw detector, the cross-sectional area of the wire rope changes, the saturation state of the wire rope is broken, and the internal magnetic field of the wire rope overflows and causes leakage. The radially mounted Hall-effect sensor scans the leakage distribution near the wire rope and determines the existence and extent of damage according to the amount of leakage [17].

According to the different scanning directions, the installation direction of the Hall-effect sensor is divided into circumferential, radial, and axial, as shown in Figure 2. Among them, the radial installation has a smaller lift-off value compared with the axial or circumferential Hall-effect sensor [18].

The practical engineering application shows that even if the defects of the same geometric shape are caused by the inclination of the wire rope, the change of the magnetization direction will also cause a completely different leakage magnetic field distribution. Figure 3a shows the wire rope and magnetic imaging scanner with the same frequency tilt, wherein the impact on the radial MFL detection degree is low, where ∆*H*_1_ is the angle of synchronous tilt between the magnetic imaging scanner and the target steel wire rope. Figure 3b shows the axis of the wire rope and magnetic imaging scanner offset, permanent magnets on the wire rope, and damage magnetization changes; in turn, ∆*θ*_1_ and ∆*H*_2_, respectively, indicate the horizontal line tilt angle of the wire rope in the flaw detector, and the offset radial distance of the wire rope in the magnetic imaging scanner. In this study, we mainly discuss the second case and its impact on MFL detection.

Figure 4 illustrates the dipole representation of the wire-rope tilt, the width of the rectangular groove is 2*b* and the depth is *h*. With the defect center as the origin and the magnetization direction as the *X*-axis, the Cartesian coordinate system *X–Y* can be established [19,20,21,22]. At this time, the magnetic field intensity *dH*_1_, and *dH*_2_ generated by the microelement on the two groove walls at the field point *P (x, y)* can be expressed as:(1)$$\left\{ \begin{array}{l} dH_{1}=\frac{\sigma msd\xi }{2\pi \mu_{0}r_{1}^{2}}\vec{r_{1}}\\ dH_{2}=\frac{\sigma msd\xi }{2\pi \mu_{0}r_{2}^{2}}\vec{r_{2}} \end{array}\right.$$

In the formula, *μ*_0_ is the vacuum permeability, *r*_1_ and *r*_2_ are the distances between the field point *P* and the positive and negative magnetic charge lines.

If the distance from the microlinear element *dξ* to the material surface is noted as *ξ*, the distance can then be expressed as:(2)$$r_{{}^{1}}^{2}={\left(x-b\right)}^{2}+{\left(y-\xi \right)}^{2},r_{{}^{2}}^{2}={\left(x+b\right)}^{2}+{\left(y-\xi \right)}^{2}$$

In the *X–Y* coordinate system, the magnetic field intensity generated by the two slot-wall microwire is expressed as:(3)$$\left\{ \begin{array}{l} dH_{1x}=\frac{\sigma ms\left(x+b\right)d\xi }{2\pi \mu_{0}\left[{\left(x+b\right)}^{2}+{\left(y-\xi \right)}^{2}\right]}\\ dH_{2x}=\frac{-\sigma ms\left(x-b\right)d\xi }{2\pi \mu_{0}\left[{\left(x-b\right)}^{2}+{\left(y-\xi \right)}^{2}\right]} \end{array}\right.$$
(4)$$\left\{ \begin{array}{l} dH_{1y}=\frac{\sigma ms\left(y-\xi \right)d\xi }{2\pi \mu_{0}\left[{\left(x+b\right)}^{2}+{\left(y-\xi \right)}^{2}\right]}\\ dH_{2y}=\frac{-\sigma ms\left(y-\xi \right)d\xi }{2\pi \mu_{0}\left[{\left(x-b\right)}^{2}+{\left(y-\xi \right)}^{2}\right]} \end{array}\right.$$
where *dH*_1*x*_ and *dH*_2*x*_ are the *X* components of the magnetic field intensities generated by the two slot-wall microstrip elements at point, and *dH*_1*y*_, *dH*_2*y*_ are the *Y* components of the magnetic field intensities produced by two slot-wall elements at point *P*. Integrating the above equation along with the depth of the slot wall *h*, the magnitude of the *X–Y* axis components *H_x_* and *H_y_* of the leakage field formed by the two slot walls at the Hall-effect sensor calibration detection point *P* (*x, y*) can be obtained as follows:(5)$$\begin{array}{c} H_{x}=\int_{-h}^{0}dH_{1X}+\int_{-h}^{0}dH_{2x}\\ =\frac{\sigma ms}{2\pi \mu_{0}}(\tan^{-1}\frac{h\left(x+b\right)}{{\left(x+b\right)}^{2}+y\left(y+h\right)}-\tan^{-1}\frac{h\left(x-b\right)}{{\left(x-b\right)}^{2}+y\left(y+h\right)})\\ H_{y}=\int_{-h}^{0}dH_{1y}+\int_{-h}^{0}dH_{2y}\\ =\frac{\sigma ms}{2\pi \mu_{0}}\ln\frac{\left[{\left(x+b\right)}^{2}+{\left(y+h\right)}^{2}\right]\left[{\left(x-b\right)}^{2}+y^{2}\right]}{\left[{\left(x-b\right)}^{2}+{\left(y+h\right)}^{2}\right]\left[{\left(x+b\right)}^{2}+y^{2}\right]} \end{array}$$

According to the principle of the magnetic dipole model, the value of *σ*_ms_ can be calculated as follows:(6)$$\sigma_{\mathrm{ms}}=\left(\frac{h/b+1}{h/\left(b\mu \right)+1}\right)H_{0}$$

In the formula, *μ* is the relative permeability of the tested material, and *b* is the external magnetic field intensity.

The magnetic field intensities *dH*_1_ and *dH*_2_ produced at the point of *P* (*x, y*) on the wall of the two grooves *dξ* are calibrated when the wire rope is running horizontally. When the wire rope vibrates, the external magnetic field *B*_0_ shifts, and the affective component of the groove wall decreases; thus, the linear infinitesimal value formed on the groove wall decreases [23]. *B*_1_ is the applied magnetic field acting on the surface of the wire rope after the deflection of *B*_0_. When the Hall-effect sensor measures the same leakage magnetic value as the calibration point *P* (*x, y*), the Hall-effect sensor needs to measure the leakage magnetic value of the moving point *P_n_*_1_ (*x_n_*_1_*, y_n_*_1_), and the Hall-effect sensor is usually designed to operate at a fixed position for the actual detection.

Therefore, to improve the magnetization effect and the leakage magnetic field of the flaw detector, it is important to build a magnetic bridge between the wire-rope damage and the Hall-effect sensor.

## 3. Mathematical Model and Simulation Model

According to the characteristics of the radial acquisition mode, the radial magnetic gathering sensor is designed to comprise two parts: a Hall-effect sensor array, and a magnetic concentrator bridge. The magnetic concentrator bridge is divided into short- and long-neck magnetic concentrator bridges. The Hall-effect sensor array is located between the short- and the long-neck magnetic bridges. The MFL signal is aggregated into the Hall-effect sensor through the short-neck magnetic concentrator bridge and returns to the wire rope through the long-neck magnetic concentrator bridge to form a closed magnetic circuit. A three-dimensional model of the radial magnetic concentrator is shown in Figure 5.

In this study, the radial magnetic concentrator sensor is designed according to the radial MFL detection equipment, and the mathematical model of the radial magnetic concentrator is established according to the established radial magnetic concentrator bridge, as shown in Figure 6.

Ignoring the edge effect of the magnetic resistance of the magnetic structure, the calculation method of the air-gap permeability between the magnetic paths of the magnetic structure is derived. The air-gap permeability *G*_1_ between the side of the sixteen magnetic concentrators is:(7)$$G_{1}=8\mu_{0}\frac{\pi {\left(N/2\right)}^{2}}{∆H}$$

In the formula listed above, *N* is the diameter of the header, ∆*H* is the air-gap length between the header and the long arm, and *μ*_0_ is the permeability of air.

According to the loop direction of the radial MFL, the radial MFL loop is simplified, and the air-gap permeance *G*_2_ between the upper side of the two concentrators in the radial loop is:(8)$$G_{2}=\mu_{0}\frac{\left[2\pi RD-8\pi {\left(\frac{N}{2}\right)}^{2}\right]}{L_{1}+L_{2}+L_{3}+M+∆H}$$
where *L*_1_, *L*_2_, and *L*_3_ are the lengths of the first, second, and third sections of the long-arm shank, respectively, *M* is the platform height, *D* is the bearing length, *R* is the inner diameter of the bush, and *T* is the bearing width.

If the ratio of the magnetic flux detected by the Hall-effect sensor to the total magnetic flux flowing through the magnetic ring is the magnetic concentration efficiency, then
(9)$$\begin{array}{c} \eta =\frac{G_{1}}{G_{1}+G_{2}}\\ =\frac{8\mu_{0}\frac{\pi {\left(N/2\right)}^{2}}{∆H}}{8\mu_{0}\frac{\pi {\left(N/2\right)}^{2}}{∆H}+\mu_{0}\frac{\left[2\pi RD-8\pi {\left(\frac{N}{2}\right)}^{2}\right]}{L_{1}+L_{2}+L_{3}+M+∆H}}\\ ={\left(1+\frac{∆H\left(\frac{RD}{N^{2}}-1\right)}{L_{1}+L_{2}+L_{3}+M+∆H}\right)}^{-1} \end{array}$$

As shown in Equation (9), to increase the value of *η*, we can increase *N*, *L*_1_, *L*_2_, *L*_3_, and *M*, or reduce ∆*H*.

To verify the influences of the variations of *L*_1_, *L*_2_, *L*_3_, and *M* on the effect of magnetic gathering, the magnetic field model of the wire-rope excitation is analyzed by numerical finite element simulations, and the detection effect which is subject to the condition of existing magnetic gathering devices is compared and analyzed. A wire rope with a length of 270 mm and a diameter of 8 mm is set as the origin of the coordinate axis, and the axial length of 8 mm and the depth of damage of 4 mm are set above the origin. The lifting-off value y is set as 6 mm. The selected magnet was an N48 NdFeB permanent magnet, the coercive force was set to 920,000 A/m, and the permanent magnetization was 1.36 T. The observation line with the length of the observation line *L*_n_ with a length of 40 mm was set at a distance of 10 mm from the origin in the *Y* direction and from −24 mm to 16 mm in the *X*-direction.

Figure 7, Figure 8, Figure 9 and Figure 10 clearly show the effect on the leakage magnetism when the parameters of *L*_1_, *L*_2_, *L*_3_, and *M* are changed. To further analyze the influence law of each parameter, the peak values of *L*_1_, *L*_2_, *L*_3_, and *M* are extracted, as shown in Figure 11. As *L*_1_ and *L*_3_ increase, the magnetic induction intensity value at the defect increases, and when *L_1_* and *L*_3_ increase to 5 mm the magnetic field inside the magnetic concentrator gradually saturates, which makes the magnetic flux detected by the Hall-effect sensor reach saturation quickly, and the growth rate of *η* decreases. Therefore, *L*_1_ and *L*_3_ are chosen to be 5 mm to make the magnetic concentrator reach the best efficiency. *L*_2_ has a slower effect on *η*, and *L*_2_ determines the axial length of the magnetic concentrator; thus, the minimum size of the flexible printed circuit (FPC) board installation should be maintained when *L*_2_ is chosen. As *M* increases, the magnetic induction intensity value at the defect increases, however, the height of *M* will affect the detection performance of the Hall-effect sensor. Excessively large *M* values will increase the lift-off value of the Hall-effect sensor. Thus, setting *M* to 4 mm is optimal, and the size of *N* needs to be matched with the width of the selected Hall-effect sensor sensing surface. Therefore, *N* should be the maximum value of the surface width of the Hall-effect sensor.

Based on the above analysis, ∆*H* is simulated in the optimal *L*_1_, *L*_2_, *L*_3_, and *M* environments, and the effect of ∆*H* on the magnetic concentration is determined by measuring the flux leakage at the center of the reference line. According to Figure 12, with the increase in ∆*H*, the defect magnetic induction intensity decreases. When ∆*H* continues to increase, the lift-off value between the long- and the short-neck magnetic concentrators is too large, and the magnetic concentrator cannot concentrate the air-gap magnetic field. Therefore, to achieve the best magnetic field aggregation effect, the lift-off value of the convex end-face between the long- and the short-neck magnetic concentrators should be kept at a small distance.

A ∆*H* of 2 mm is used as the air-gap distance of the convex platform of the magnetic concentrator to verify the magnetic concentrator effect on the inclined wire rope. The swing angle of the wire rope is determined by the inner hole radius of the flaw detector. The inner lining radius of the flaw detector was 22 mm. Therefore, the swing angle *θ* of the wire-rope axis and the horizontal axis was set to values in the range of 0.5–3° and the angle increment to 0.5°. The MFL induction intensity with the defect axial length of 8 mm and the depth of 4 mm was measured at position *L*_n_.

Figure 13a shows the magnetic induction intensity value of the magnetic concentrator installed in the angular swing range of 0.5–3°. According to the results shown in Figure 13a, with the increase in the swing angle, the peak value of the magnetic induction intensity does not decrease considerably, and the peak value at the damage location remains at approximately 426 mT. Figure 13b shows the magnetic induction intensity value in the case in which there was no magnetic concentrator in the angular swing range of 0.5–3°. According to the results shown in Figure 13b, with the increase in the swing angle, the peak value of magnetic induction intensity decreases gradually, the peak value at the damage is maintained at approximately 236 mT, and the peak slows down at a low rate.

## 4. Design of Wire-Rope Detector

### 4.1. Circuit Design of Sensor

The FPC plate was used as the carrier of the sensor in synergy with the magnetic concentrator, and the Hall-effect sensor was uniformly distributed on the FPC plate. The FPC is easy to bend, but the radial lift-off value can be controlled and can be maintained at its minimum. When the FPC plate is bent with the inner cavity of the flaw detector, the Hall-effect sensor array is arranged into a radial circular array, and the Hall-effect sensor array corresponds to the convex position of the magnetic concentrator, as shown in Figure 14.

The damage to the wire rope may occur at any position in its circumferential direction. To ensure the comprehensive detection of wire-rope damage, multiple Hall-effect sensors should be arranged uniformly and densely along the circumference of the same section of the wire rope. The specific number should meet the following condition:(10)$$N_{h}=CINT\left[\frac{S_{p}}{S_{s}}\right]$$

In the formula listed above, *S_p_* is the perimeter of the ring surface of the Hall-effect sensor, *S_s_* is the circumferential coverage area of the single Hall-effect sensor, and *CINT* [•] is the round-off integral function. The Hall-effect sensors of the THS119 series were used in the experiment. According to the number of circumferential arrangements of the existing flawed detector, the maximum outer contour radius of the antiwear lining of the flawed detector was 22 mm, and the inner arc radius of the concentrator was 27 mm. When the FPC plate was installed in the radius range of 22–27, the *S_p_* was in the range of 139–170, and the circumferential coverage range of the THS119 series Hall-effect sensors was in the range of 17–20. According to the actual measurements, the *S_s_* value was 20, and the input was equal to 10. The theoretical value of *N_h_* was in the range of 7–8.5. Thus, *N_h_* was assumed to be equal to eight.

As shown in Figure 15, pins 1 and 3 are input, namely, they are the power pins, while pins 2 and 4 are the differential outputs (The red letters 1, 2, 3 and 4 in Figure 15 indicate the pin locations of the Hall-effect sensors.). When the power is supplied to the Hall-effect sensor, a constant current in the range of 5 or 10 mA is achieved when the Hall-effect sensor works in a linear region. According to *N_h_* = 8, the FPC board, which is unfolded in the plane shown in Figure 16, is equipped with eight Hall-effect sensors, each of which was 20 mm apart. These sensors were powered in series by a constant current source that simultaneously supplied two Hall-effect sensors, such as A_1_ and A_2_, as shown in Figure 15. In the A_1_ and A_2_ sensors, the positive electrode of the constant current source was connected to the first pin of A_1_, and the negative electrode was connected to the third pin of A_2_. At the same time, the third pin of A_1_ was connected to the first pin of A_2_.

### 4.2. Mounting of FPC Boards

The FPC plate was installed in the wire-rope detector in a curved way, the working surface of the Hall-effect sensor was parallel to the surface of the wire rope, and the lift-off value was controlled in the minimum range. The wire-rope detector consisted of upper and lower parts with permanent magnets made of neodymium boron as the excitation source. Additionally, the left and right permanent magnets were wrapped around the wire rope once, thus forming a closed magnetic circuit with the wire rope. The FPC plate is wrapped around the liner, which is used to isolate the wire rope and to support the FPC and the magnetic concentrator. The magnetic concentrator was mounted in the gap between the liner and the armature, and the tab of the magnetic concentrator corresponded to the position of the Hall-effect sensor on the FPC board at the end (near the tab shank), as shown in Figure 17. There are two types of samplings: temporal and spatial. In the process of using equally spaced pulse signals to control the damage signal sampling to ensure that no slippage was generated between the roller and the wire rope, the elastic damping member was added to the wire-rope detector.

## 5. Experiments

### 5.1. Experimental Setting

To verify the radial magnetic concentration effect of the wire rope, the following experiments are designed:Wire rope and the axis of the flawed detector overlap.We selected 7 × 6 × 19 FS wire ropes prefabricated with external damage with a fixed number of broken wire roots, a starting value of the length of the broken wire of 3 mm, and regular production damage with a length gain of 2 mm. The detailed parameters for the damage to the prefabricated wire rope are shown in Figure 18. To reduce the influence of adjacent damage, it is recommended that the axial distance between the prefabricated damage (300 mm) is greater than the length of the flaw detector (270 mm). Figure 19 shows the experiment associated with the installation of the magnetic concentrator in which the implementation of the long- and the short-neck magnetic concentrator is shown. The sensor with the concentrator removed is used to detect damage (conventional detection method). Figure 20 clearly illustrates the detector composition for the conventional detection method. The FPC board was wound in the gap of the three-dimensional printed liner, and the end of the FPC board was welded with an external circuit quick plug to realize the power input and the output of the signal outside the detector.Axis deviation of wire rope and flaw detector.The tilt experiment was performed using the magnetic concentrator method and the conventional detection method, where the magnetic concentrator method used the structure in Figure 19 and the conventional detection method was the device in Figure 20. This part of the experiment was conducted on the non-destructive testing platform of the wire rope. By lifting the wheel shaft of the experimental platform, the inclination angle of the wire rope relative to the flaw detector was adjusted. As shown in Figure 21, the red-dotted line shows the horizontal position of the bearing seat, the yellow-dotted line shows the horizontal position of the wheel shaft, the blue-dotted line shows the adjustment height of the wheel shaft, the double-dotted yellow line shows the horizontal position of the wire rope, and the rough yellow solid line shows the inclined angle of the wire rope after adjustment. A closed-loop wire rope wound on the detection platform was selected, and the damage with 19 broken wires and 10 mm broken wires were produced at an interval of 30 mm. Taking the running direction V of the wire rope as the positive direction of the *X*-axis, the wire rope was set to deflect by an angle θ in the positive direction of the *X*-axis. On the detection platform, the vertical position of the wheel disc was changed by adjusting the wheel shaft ∆G, so as to realize the change of the wire-rope inclination. In this case, θ was in the range of 0.5–3.0°, and the change angle increment was 0.5°.

According to the damage location, the broken line signal was detected many times. Firstly, the amplifier circuit was used to amplify the differential voltage signal output of the Hall-effect sensor. The data acquisition card was used to collect signals, and the collected signals were saved on the computer. The USB 5633 data acquisition card of Beijing Altay Technology Development Company Ltd. (Beijing, China) was used. The highest acquisition frequency of this data acquisition card can be set to 260 kHz.

### 5.2. Data Analysis

In the signal processing section, the acquired damage signals were converted first and were then saved in an Excel table in a text format. The data in the table were plotted using MATLAB (version 2020b, MathWorks, Natick, MA, USA). The wire rope level controls experimental data for Figure 22. Experiments measured damage from large to small values, the horizontal coordinate in the graph is the detection time in milliseconds (ms), and the vertical coordinate is the detection voltage amplitude in volts (V).

In the wire-rope tilt experiment shown in Figure 23, the traditional leakage detection method with the wire-rope running tilt angle becomes larger. In the detection of the same damage, the amplitude of the damage signal is gradually reduced. Furthermore, because the wire rope tilted, the detected damage signal was mixed with more odd signals. In the magnetic concentrator detection signal, the amplitude of the damage signal increased, thus indicating that the air-gap magnetic field at the damage is aggregated by the magnetic concentrator.

To further explore the relationship between the number of broken wires and signals, it is necessary to conduct in-depth processing of the obtained signals. The peak value, valley value, and peak and valley areas of the two groups of the experimentally interrupted rope signals were derived, and the derived signals were properly counted. These included the axial overlap magnetic concentration experiment of the wire rope (Table 1) and the inclined magnetic concentration of the wire rope (Table 2).

Given that there are errors in each damage detection, the average percentage increase in the peak, valley, and peak and valley areas of the damage signal at the same experimental conditions should be regarded as the analysis object. Table 3 shows the overall improved performance of the concentrator on the peak-to-peak value and the waveform area at two experimental conditions.

From the horizontal experiment of the wire rope in Table 3, the magnetic concentrator increases the peak value by 46% and the waveform area by 145%. The magnetic concentrator generates a polymerization magnetic field at the damage sites of the wire rope with uniform length changes and improves the MFL collection efficiency of the Hall-effect sensor. In the tilt experiment, the peak value and waveform areas were increased by 130% and 177%, respectively, and the average peak value and waveform area were maintained at 1300 mV and 7200 mV, respectively. The signal-to-noise ratio (SNR) is the main characterization parameter of the quality of the detected signal, which reflects the ratio of the effective work of the signal and the noise in the original signal. Table 3 clearly shows that the SNR increased by 7.29 dB (21.89 dB > 14.60 dB) after the installation of the concentrator in the wire-rope level experiment, and by 7.72 dB (25.12 dB > 17.40 dB) after the installation of the concentrator in the wire-rope tilt experiment. It is clearly shown that when the magnetic concentrator is installed, the Hall-effect sensor can still effectively collect the leakage magnetic field at the damage location when the wire rope is tilted, and the leakage magnetic field at the damage location remains stable so that the problem of wire-rope tilt jitter is solved to a certain extent when using the MFL detection.

## 6. Conclusions

To solve the influence of wire-rope inclination on damage magnetic field leakage, this study proposed a radial magnetic concentration method of the wire rope based on the magnetic circuit law. To construct a radial magnetic concentrator bridge, a magnetic circuit was established inside the flaw detector, and a sensor based on radial magnetic concentration was designed. The research contents of this study mainly include:The analysis of installation methods of Hall-effect sensors in typical MFL detection arrangements, and the comparison of the characteristics of different installation methods. According to the damage detection method and principle of radial installation, the radial detection was selected as the best strategy for the construction of magnetic concentrator bridges;The inclined vibration type of the wire rope in the flawed detector case was clarified, and the two-dimensional magnetic dipole model was established at the damage point to determine the basic variation law of MFL;The establishment of a radial magnetic concentrator bridge, an analysis of the effect of magnetic concentrator size and air-gap distance on the leakage field, and the selection of a suitable structure size to make the efficiency optimal;The horizontal and tilt detection experiments of wire rope based on a radial magnetic concentrator bridge were designed. The horizontal experiment adopted seven types of wire-rope damage assessments and the tilt experiment adopted a single type of damage. The control variable method of different tilt angles was used for evaluation;Compared with the experimental results of the two groups, it was shown that the radial magnetic concentrator can be used to obtain more disrupted line signals. From the horizontal experimental results, the peak value of the radial magnetic concentrator for different lengths of damage increased by 46%, and the waveform area increased by 145%. The magnetic concentrator had an efficient magnetic effect on the axial length change in the horizontal test experiment. According to the tilt test results, the peak value of the radial magnetic concentrator can be increased by 130% and the waveform area can be increased by 177%. In the tilt test, the magnetic concentrator not only polymerized the leakage magnetic field of the damage but also stabilized the leakage magnetic field of the damage.

## Figures and Tables

**Figure 1 sensors-22-03654-f001:**
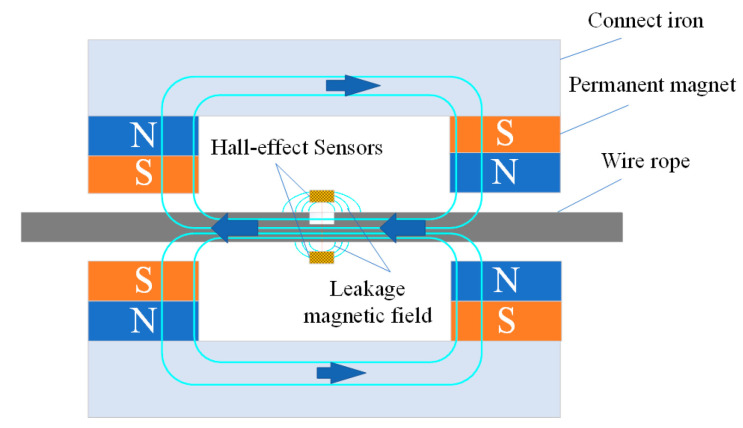
Principle diagram of MFL detection.

**Figure 2 sensors-22-03654-f002:**
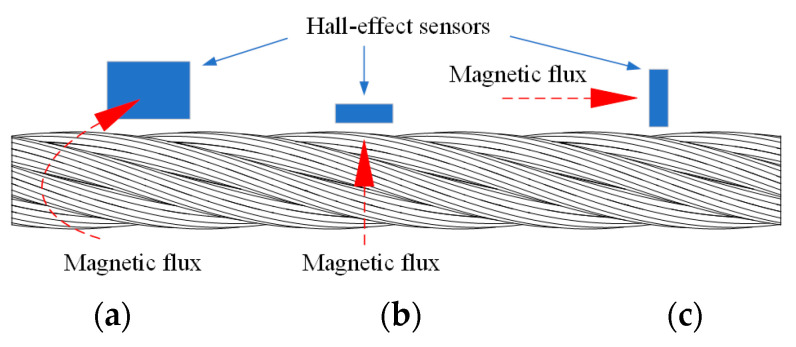
Schematic diagram of circumferential, radial, and axial positions of Hall-effect sensor: (**a**) Circumferential installation; (**b**) Radial installation; (**c**) Axial installation.

**Figure 3 sensors-22-03654-f003:**
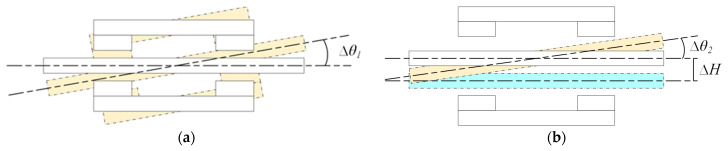
Tilt-type diagram of wire rope: (**a**) Tilting at the same frequency; (**b**) Axis offset tilt.

**Figure 4 sensors-22-03654-f004:**
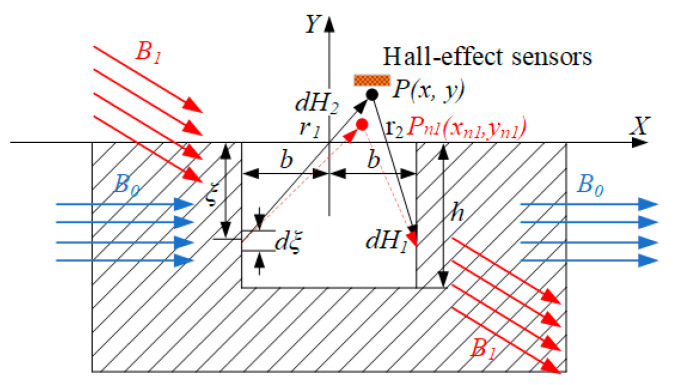
Mathematical model of tilt magnetization.

**Figure 5 sensors-22-03654-f005:**
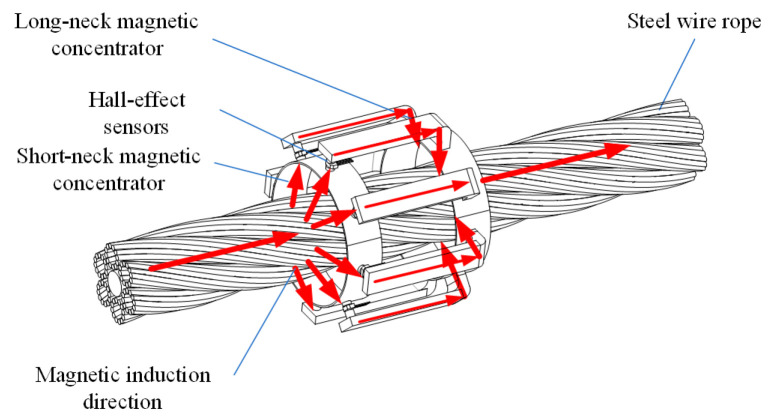
Three-dimensional model of radial magnetic concentrator.

**Figure 6 sensors-22-03654-f006:**
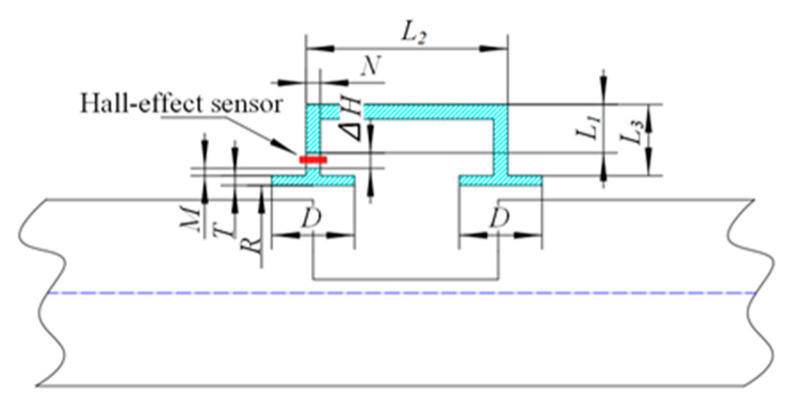
Mathematical model of radial magnetic concentrator.

**Figure 7 sensors-22-03654-f007:**
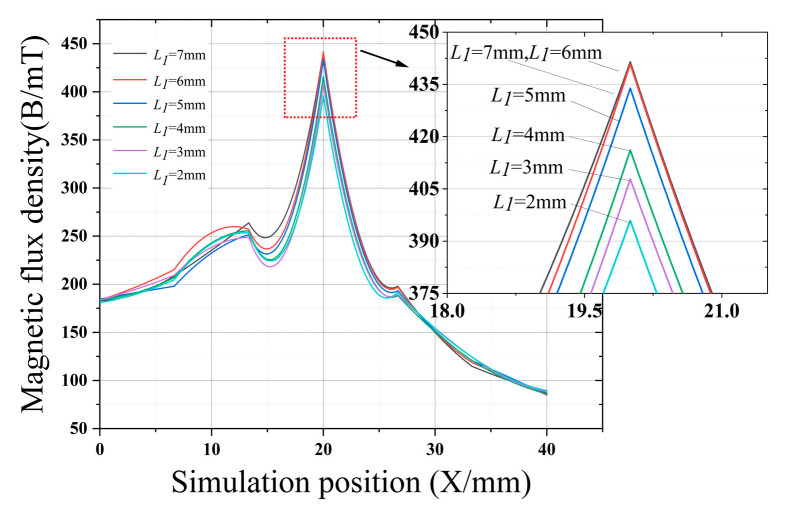
Effect of *L*_1_ on the magnetic concentrator.

**Figure 8 sensors-22-03654-f008:**
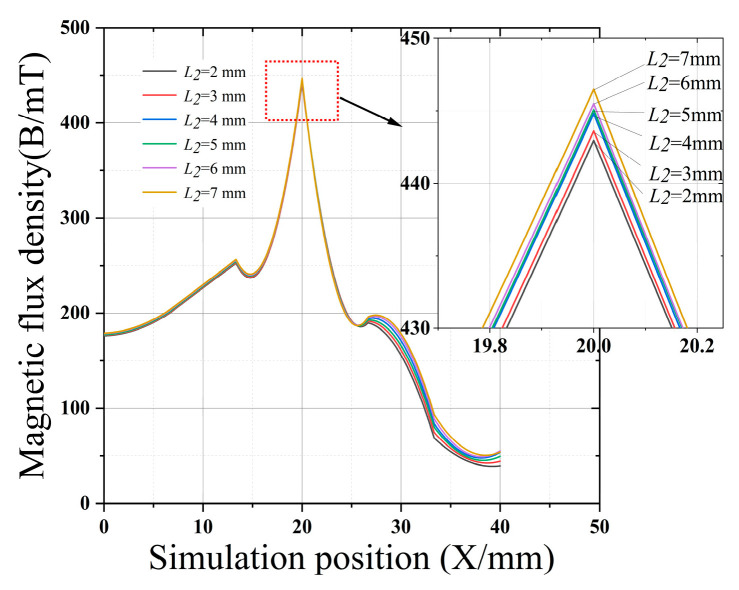
Effect of *L*_2_ on the magnetic concentrator.

**Figure 9 sensors-22-03654-f009:**
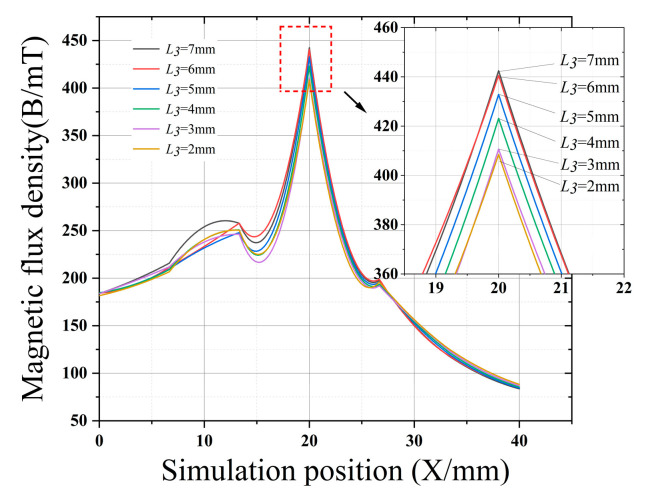
Effect of *L*_3_ on the magnetic concentrator.

**Figure 10 sensors-22-03654-f010:**
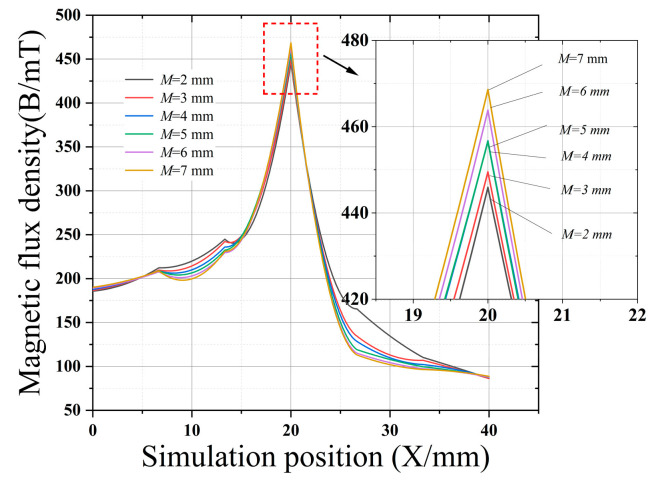
Effect of *M* on the magnetic concentrator.

**Figure 11 sensors-22-03654-f011:**
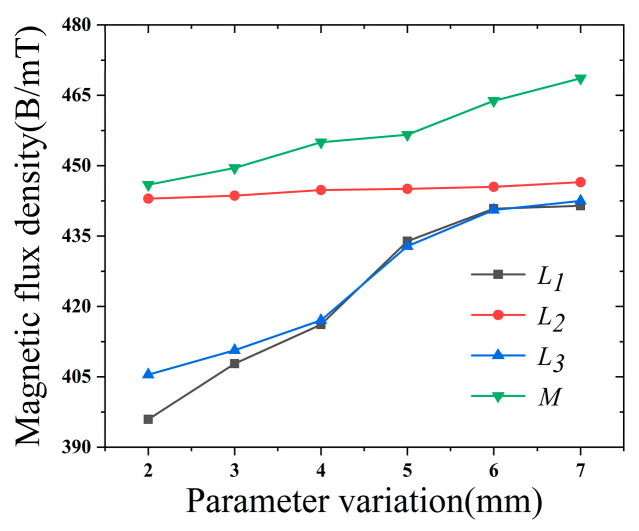
Leakage density at different parameters.

**Figure 12 sensors-22-03654-f012:**
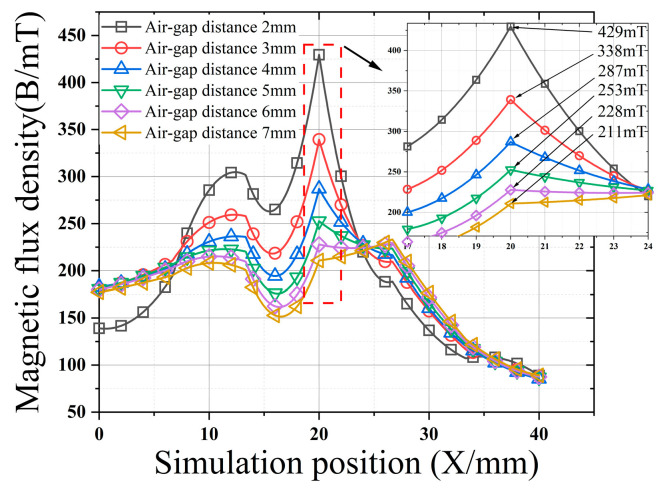
Effect of ∆*H* on the magnetic concentrator.

**Figure 13 sensors-22-03654-f013:**
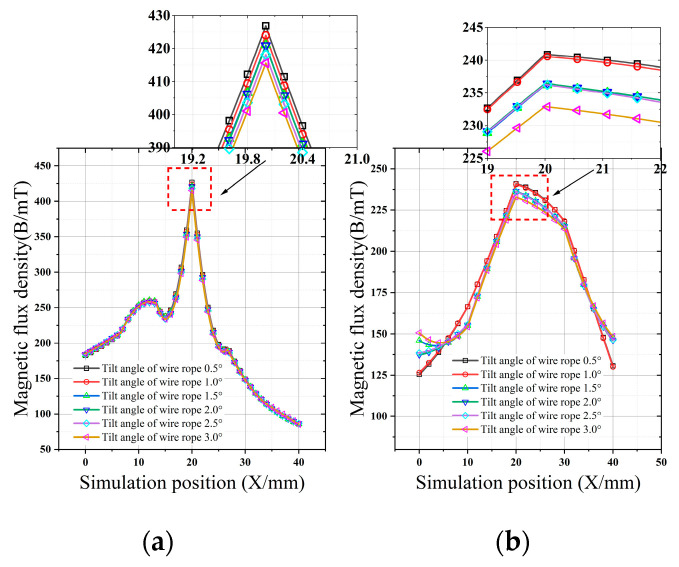
Comparison of magnetic induction intensity after installing the magnetic concentrator. (**a**) The magnetic induction intensity when the wire rope is inclined. (**b**) Inclined magnetic induction intensity of wire rope without concentrator.

**Figure 14 sensors-22-03654-f014:**
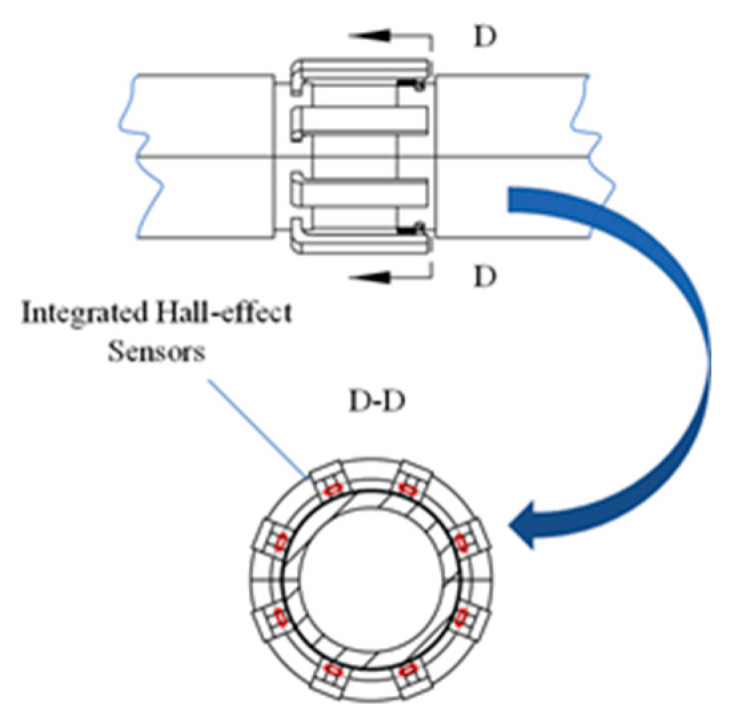
Installation diagram of Hall-effect sensor and radial magnetic concentrator.

**Figure 15 sensors-22-03654-f015:**
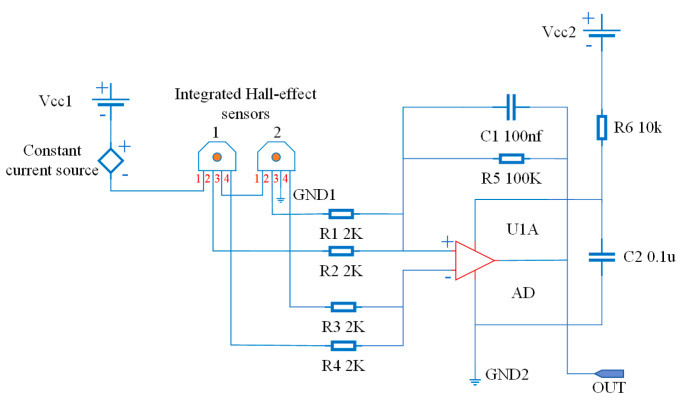
Circuit diagram of integrated Hall-effect sensor.

**Figure 16 sensors-22-03654-f016:**
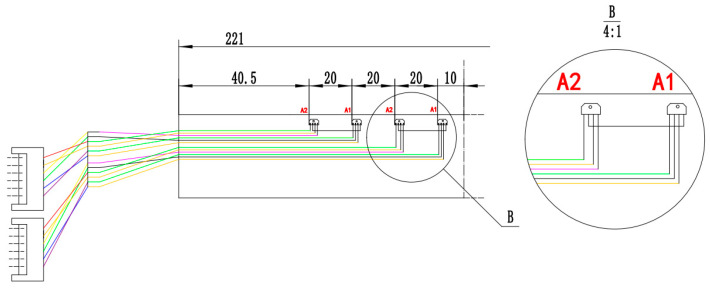
FPC circuit diagram.

**Figure 17 sensors-22-03654-f017:**
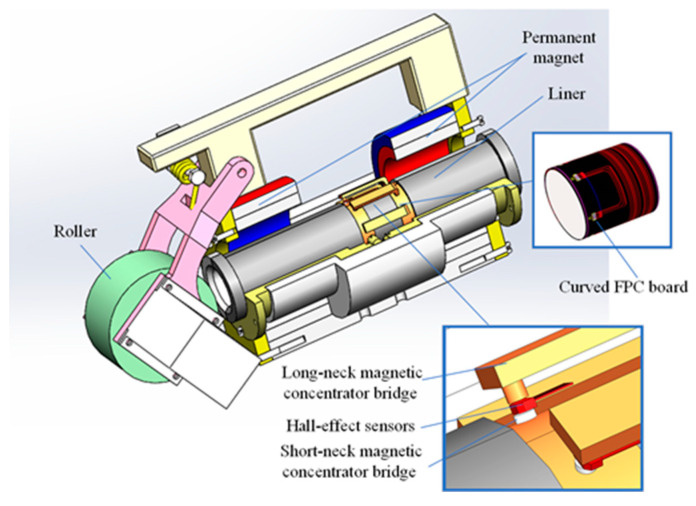
Installation of magnetic concentrator and FPC board.

**Figure 18 sensors-22-03654-f018:**
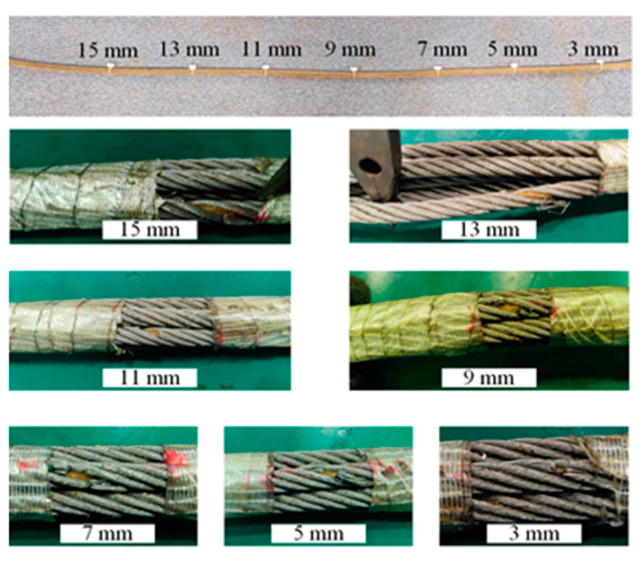
Axial variation damage of steel wire rope.

**Figure 19 sensors-22-03654-f019:**
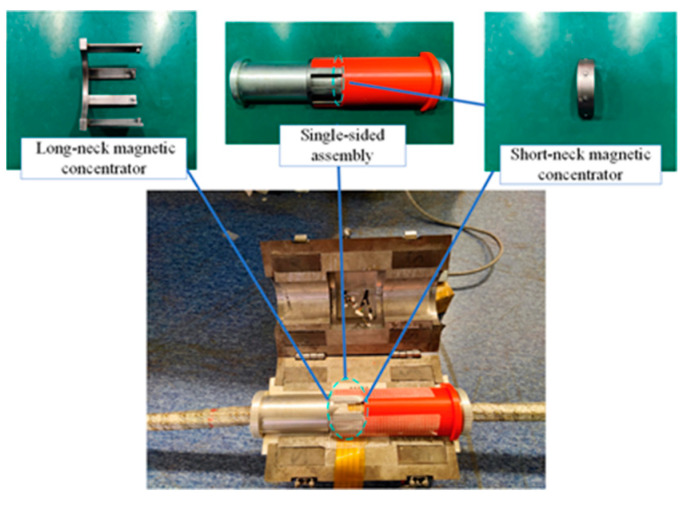
Radial magnetic concentration experiment.

**Figure 20 sensors-22-03654-f020:**
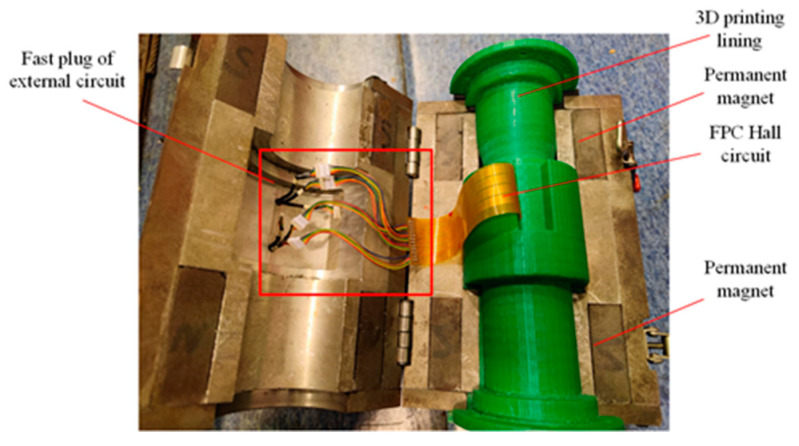
Non-aggregation magnetic experiment.

**Figure 21 sensors-22-03654-f021:**
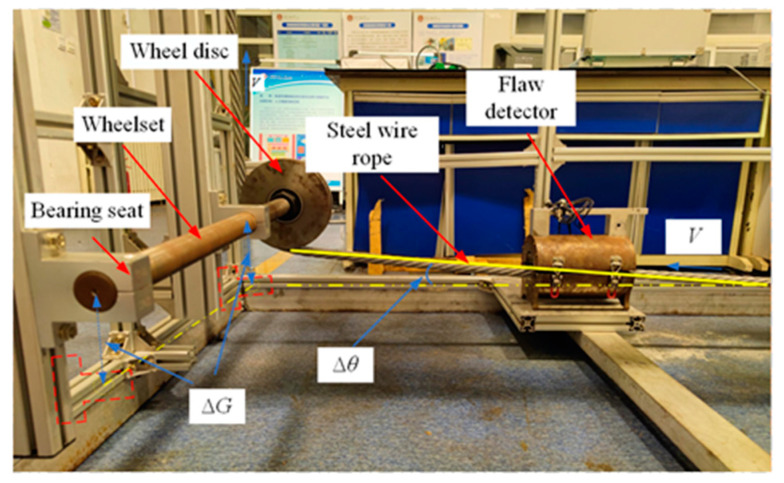
Tilting experiment of steel wire rope.

**Figure 22 sensors-22-03654-f022:**
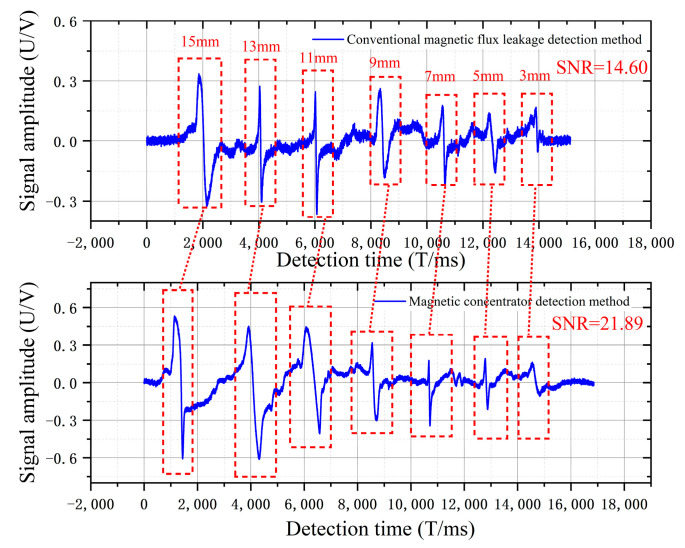
Magnetic contrast experiments with different damage lengths.

**Figure 23 sensors-22-03654-f023:**
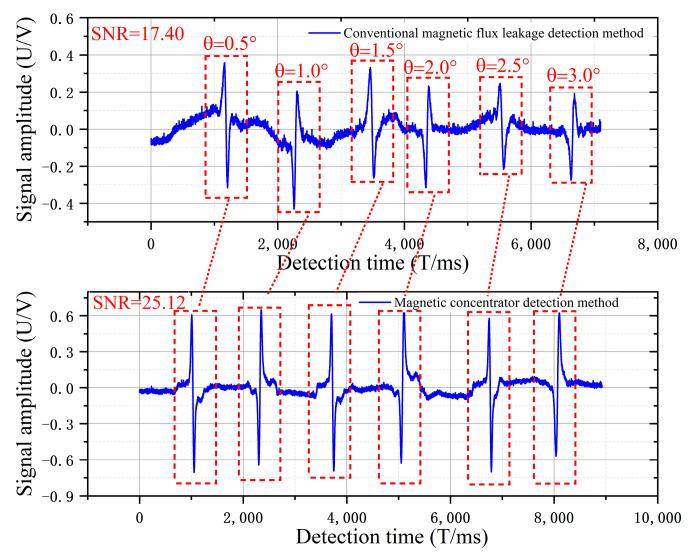
Comparative experiments of magnetic concentration at different tilt angles.

**Table 1 sensors-22-03654-t001:** Signal characteristic statistics.

	Broken Distance	15 mm	13 mm	11 mm	9 mm	7 mm	5 mm	3 mm
Orthodox Method	Peak (mV)	335	273	246	263	177	140	168
	Valley (mV)	−327	−304	−366	−185	−224	−159	−55
	Peak-to-peak (mV)	662	577	612	448	401	299	223
	Peak-area (mV * ms)	8690	1389	1307	4095	1853	2059	1679
	Valley-area (mV * ms)	−9569	−5404	−4167	−2707	−2611	−1810	−914
	Waveform (mV * ms)	18,259	6793	5474	6802	4464	3869	2593
Magnetic Concentrator	Peak (mV)	532	450	446	318	178	193	163
	Valley (mV)	−609	−612	−408	−304	−345	−213	−109
	Peak-to-peak (mV)	1141	1062	854	622	523	406	272
	Peak-area (mV * ms)	15,772	14,668	13,731	3166	429	1029	2762
	Valley-area (mV * ms)	−25,916	−27,009	−6820	−5373	−3357	−1875	−2795
	Waveform (mV * ms)	41,688	41,677	20,551	8539	3786	2904	5557

**Table 2 sensors-22-03654-t002:** Signal characteristic statistics.

	Bevel Angle	0.5°	1.0°	1.5°	2.0°	2.5°	3.0°
Orthodox Method	Peak (mV)	358	205	332	232	248	192
	Valley (mV)	−317	−431	−264	−316	−216	−276
	Peak-to-peak (mV)	675	636	596	548	464	468
	Peak-area (mV * ms)	1442	1415	2068	1010	930	990
	Valley-area (mV * ms)	−1511	−1500	−1274	−1397	−1590	−996
	Waveform (mV * ms)	2953	2915	3342	2407	2520	1986
Magnetic Concentrator	Peak (mV)	607	648	614	658	577	633
	Valley (mV)	−707	−644	−695	−629	−700	−567
	Peak-to-peak (mV)	1314	1292	1309	1287	1277	1200
	Peak-area (mV * ms)	3025	4075	3055	5777	3227	4374
	Valley-area (mV * ms)	−4186	−2548	−4338	−2874	−3140	−2728
	Waveform (mV * ms)	7211	6623	7393	8651	6367	7102

**Table 3 sensors-22-03654-t003:** Signal characteristic statistics.

Predictable Pattern	Average Value	Proportion	SNR of Conventional Method/dB	SNR of Magnetic Concentration Method/dB
Level detection of wire rope	Peak-to-peak (mV)	46.14%	14.60	21.89
	Waveform (mV * ms)	145.28%		
Tilt detection of wire rope	Peak-to-peak (mV)	130.65%	17.40	25.12
	Waveform (mV * ms)	177.05%		

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
