# Peer review of "Enhancing Wire-Rope Damage Signals Based on a Radial Magnetic Concentrator Bridge Circuit"

_sensors, 2022, doi:10.3390/s22103654_

Round 1
Reviewer 1 Report
The authors attempted in the manuscript, to present a novel wire‐rope damage signals enhanced method. It uses a radial magnetic concentrator bridge to improve the detection effect of wire rope. The work is reasonably good. The paper is well structured. The submitted manuscript can be accepted for publication in this journal. The following corrections are recommended:
- Figure 16. must be illustrated in a high quality.
- The math equations need to be supported by valid reference. Please double check this issue.
- Presentation of the results (Figures 11, 12 and 13) is not clear. I recommend to use high resolution pictures.
Reviewer 2 Report
The authors study a radial magnetic flux concentrator used in experiments for nondestructive testing of wire ropes. They present a theoretical analysis of this method, as well as transient magnetic field simulations. Experimental data on wire ropes with artificial damges are also shown. The paper presents interesting work and deserves publication. I have two minor comments, the authors might want to address:
In Figs. 22 and 23 the authors compare data taken with the 'traditional leakage detection method' and also with their flux concentrator. It is not quite clear if in both cases one and the same system was used, with the exception that in the 'traditional case' the flux concentrator was simply removed, or if different equipment was used. The authors state on page 12 that 'a control group experiment was also set up for the same damage type experiment without the magnetic concentrator', but this statement needs clarification.
The authors could show that by using the flux concentrator, there was an improvement in the peak-to-peak value and the waveform area of the measured signal. Nevertheless, the signal amplitude alone is not necessarily determing a potential advantage of a particular method; the signal-to-noise ratio also needs to be considered: In Figs. 22 and 23, there indeed seems to be an improvement in the signal-to-noise of the defects. The figure given by the authors of an improvement in the signal amplitude (peak value) of 46% (see line 455) corresponds to an improvements of ~ 3dB, which, naively, is not so much. It would thus be helpful for potential readers of the paper if the authors could add a few words about the expected improvement in the signal-to-noise ratio compared to traditional methods.
